# Porcine Deltacoronavirus Utilizes Sialic Acid as an Attachment Receptor and Trypsin Can Influence the Binding Activity

**DOI:** 10.3390/v13122442

**Published:** 2021-12-06

**Authors:** Yixin Yuan, Shaopo Zu, Yunfei Zhang, Fujie Zhao, Xiaohui Jin, Hui Hu

**Affiliations:** 1College of Veterinary Medicine, Henan Agricultural University, Zhengzhou 450046, China; yuanyixin94@163.com (Y.Y.); zushaopo@163.com (S.Z.); a15238090193@163.com (Y.Z.); zfj216403@163.com (F.Z.); jinxiaohui2020@126.com (X.J.); 2Key Laboratory for Animal-Derived Food Safety of Henan Province, Zhengzhou 450046, China

**Keywords:** porcine deltacoronavirus, receptor, carbohydrates, sialic acid, trypsin

## Abstract

Porcine deltacoronavirus (PDCoV) is a novel coronavirus that causes diarrhea in nursing piglets. Studies showed that PDCoV uses porcine aminopeptidase N (pAPN) as an entry receptor, but the infection of pAPN-knockout cells or pigs with PDCoV revealed that pAPN might be not a critical functional receptor, implying there exists an unidentified receptor involved in PDCoV infection. Herein, we report that sialic acid (SA) can act as an attachment receptor for PDCoV invasion and facilitate its infection. We first demonstrated that the carbohydrates destroyed on the cell membrane using NaIO_4_ can alleviate the susceptibility of cells to PDCoV. Further study showed that the removal of SA, a typical cell-surface carbohydrate, could influence the PDCoV infectivity to the cells significantly, suggesting that SA was involved in the infection. The results of plaque assay and Western blotting revealed that SA promoted PDCoV infection by increasing the number of viruses binding to SA on the cell surface during the adsorption phase, which was also confirmed by atomic force microscopy at the microscopic level. In in vivo experiments, we found that the distribution levels of PDCoV and SA were closely relevant in the swine intestine, which contains huge amount of trypsin. We further confirmed that SA-binding capacity to PDCoV is related to the pre-treatment of PDCoV with trypsin. In conclusion, SA is a novel attachment receptor for PDCoV infection to enhance its attachment to cells, which is dependent on the pre-treatment of trypsin on PDCoV. This study paves the way for dissecting the mechanisms of PDCoV–host interactions and provides new strategies to control PDCoV infection.

## 1. Introduction

Porcine deltacoronavirus (PDCoV) is a newly emergent coronavirus that is classified in the Deltacoronavirus genus of the Coronaviridae family [1,2]. PDCoV primarily causes diarrhea, vomiting, dehydration, and death in sick piglets, and these clinical signs are similar to those of porcine epidemic diarrhea virus (PEDV) and porcine transmissible gastroenteritis virus (TGEV) [3]. PDCoV was first reported in Hong Kong, China, in 2012, and the first PDCoV OH-FD22 strain was isolated on cell lines from the intestinal contents of diarrheic pigs in 2015 in the United States [4]. Its prevalence was subsequently reported in many countries, leading to huge economic losses for the pig industry. PDCoV can infect cells of many species, including pigs, human, calves, and chickens in vitro [5,6]. It also causes self-limiting infection in chickens [7,8], posing a considerable threat to animals and human health. Recent reports showed that PDCoV was identified in plasma samples of three Haitian children with acute febrile, suggesting that PDCoV may have the ability to infect humans [9]. Nevertheless, the lack of related vaccines or drugs poses a considerable risk to public health security worldwide. Therefore, a better understanding of PDCoV infection mechanism can help to develop drugs that inhibit virus infection and transmission.

The glycocalyx is a polysaccharide-protein complex embedded in the plasma membrane. The binding of some viruses to host cell receptors is prevented by the glycocalyx on the cell surface [10]. However, some viruses, such as influenza virus, herpes simplex virus, and partial coronaviruses, have evolved to initially attach to glycan receptors mediated by the glycan-binding domains on viral membrane proteins to promote the binding of the virus to cell receptors, which facilitate membrane fusionor endocytosis and viron internalization [11]. Sialic acid (SA) is located on the ends of glycans found in glycolipids and glycoproteins [12]. Studies have suggested that several coronaviruses take heparan sulfate (HS) or SA as their glycan receptors. Among them, severe acute respiratory syndrome coronavirus 2 (SARS-CoV-2) and avian infectious bronchitis virus (IBV) take HS as glycan receptor [11,13], while Middle East respiratory syndrome coronavirus (MERS-CoV), feline enteric coronavirus (FECV), and TGEV use SA as glycan receptor [14,15,16]. A variety of coronaviruses that bind to SA have hemagglutinating activity, such as bovine coronavirus (BCoV), human coronavirus OC43 (HCoV-OC43), and hemagglutinating encephalomyelitis virus (HEV) [17]. Our previous studies confirmed that PDCoV could agglutinate rabbit erythrocytes, and the inhibition of SA on the surface of rabbit erythrocytes using neuraminidase (NA) could inhibit hemagglutination [18]. Therefore, we deduce that SA may be involved in the infection of PDCoV, but the exact mechanism is still unknown.

In this study, the role of SA was investigated during PDCoV infection, and the underlying mechanism was explored. We found that both sodium metaperiodate (NaIO_4_) and NA pretreatment could attenuate the infection of PDCoV. We further confirmed that SA could promote the attachment ability of PDCoV, and the binding activity of SA seems to be related to its enterophilic nature in the PDCoV-infected piglets. We also discovered that the SA-binding activity of PDCoV was trypsin-depended, which is consistent with the results of the erythrocyte hemagglutination assay obtained earlier in our laboratory. These findings indicated that cellular SA may serve as a co-factor for PDCoV infection and emphasize the important role of trypsin in the binding of PDCoV to cellular SA.

## 2. Materials and Methods

### 2.1. Cells and Virus

Swine testicular (ST) and LLC porcine kidney (LLC-PK1) cells were purchased from the Institute of China Veterinary Medicine Inspection. ST cells were cultured in Dulbecco’s Modified Eagle’s Medium (DMEM, Gibco, Carlsbad, CA, USA) containing 10% fetal bovine serum (FBS, Gibco, Carlsbad, CA, USA) at 37 °C in a humidified atmosphere with 5% CO_2_. LLC-PK1 cells were cultured in Minimum Essential Mmedium (MEM, Gibco, Carlsbad, CA, USA) containing 5% FBS, 1% nonessential amino acids (NEAA, Gibco, Carlsbad, CA, USA), 1% antibiotic-antimycotic (Gibco, Carlsbad, CA, USA), and 1% HEPES (Gibco, Carlsbad, CA, USA). PDCoV HNZK-02 strain (GenBank: MH708123.1) and TGEV HN-2012 strain were isolated and identified by our laboratory. VSV-GFP was provided by Professor Zhen-Hua Zheng from the Wuhan Institute of Virology, Chinese Academy of Sciences. TGEV and VSV-GFP were propagated on ST cells with maintenance medium (DMEM supplemented with 1% antibiotic-antimycotic). The maintenance medium for PDCoV propagation on LLC-PK1 cells was MEM supplemented with 1% antibiotic-antimycotic solutions, 1% NEAA, 1% HEPES, and 5 μg/mL trypsin (Gibco, Carlsbad, CA, USA). TGEV and VSV-GFP were propagated in LLC-PK1 cell with maintenance medium (MEM supplemented with 1% antibiotic-antimycotic solutions, 1% NEAA, 1% HEPES).

### 2.2. Cytotoxicity Assay

The cytotoxicity of chemicals used in this study to the ST and LLC-PK1 cells were determined using the Cell Counting Kit-8 (CCK-8, Solarbio, Beijing, China) assay. NaIO_4_ was dissolved in phosphate buffered saline (PBS, Solarbio, Beijing, China), and stored at 4 °C and protected from light. NA (Sigma, Burlington, MA, USA) was dissolved in DMEM and stored at −20 °C. Cells were seeded into 96-well plates and grown to 100% confluence after 24 h. After washing three times with Dulbecco’s phosphate buffered saline (DPBS, Solarbio, Beijing, China), cells were pre-treated with NaIO_4_ at the indicated concentrations (0, 0.2, 1, 5, 10, 20, and 50 mM) for 30 min at 4 °C or pre-treated with different concentrations of NA (0, 0.0625, 0.25, 1, 4, 16, and 64 U) for 1 h at 37 °C. Cells were washed twice with DPBS and then cultured with DMEM for 1 h at 37 °C. The inoculum was removed, and the cells were washed twice with DPBS. The maintenance medium was added to the cells, and the plates were incubated for 24 h at 37 °C. CCK-8 assay was then carried out to evaluate cell viability. The CC_50_ was calculated using GraphPad Prism software.

### 2.3. Cell Infection and Treatment

ST cell monolayers were grown in 24-well plates. After washing three times with DPBS, the cells were pre-treated with NaIO_4_ at the indicated concentrations (0.2, 1, and 5 mM) for 30 min at 4 °C or pre-treated with NA at a concentration of 0.0625, 0.25, 1, and 4 U for 1 h at 37 °C. After the pretreatment, cells were washed twice with serum-free DMEM followed by virus infection at an MOI of 0.1 for 1 h at 37 °C. The infected cells were washed twice with DPBS and replenished with the maintenance medium. Samples were collected after 24 h post-inoculation (hpi). The viral binding and infectivity assays were carried out using quantitative real-time reverse-transcription PCR (RT-qPCR) and tissue culture infectious dose 50 (TCID_50_).

### 2.4. RNA Extraction and RT-qPCR

Total RNA was isolated from the cells with Trizol Reagent, and reverse transcription (RT) was conducted using the Reverse Transcription Kit (Vazyme, Nanjing, China). RT-qPCR was performed in triplicate with SYBR Premix Ex Taq (Vazyme, Nanjing, China) according to the manufacturer’s instructions. The primers targeting PDCoV S gene (PDCoV-S-F: 5′-CGTTAACCTCTTCTCACCACTT-3′ and PDCoV-S-R:5′-GCTGAGAGTCTGGTTGGTTATT-3′) were designed based on the sequence of the USA/Iowa136/2015 strain (GenBank accession no. KX022602). Data were normalized to the level of β-actin expression in each sample and are expressed as fold differences between control and treated cells using the 2^−ΔΔCT^ method.

### 2.5. TCID_50_ Assay

TCID_50_ assay was performed to assess viral titration. Confluent LLC-PK1 cell monolayers in 96-well cell culture plates were inoculated with ten-fold serially diluted viruses (100 μL/well) at 37 °C for 1 h. The excess viral inoculum was removed by washing with PBS. Then 200 μL of maintenance medium supplemented with 5 μg/mL trypsin was added to each well, and cells were cultured for another 3–5 days. The cytopathic effect was recorded daily, and virus titers were calculated using the Reed–Muench method and recorded as TCID_50_/100 μL.

### 2.6. Immunocytofluorescence Assay

For immunocytofluorescence staining, cells were washed twice with PBS and fixed with 4% formaldehyde (Merck, Darmstadt, Germany) in PBS, followed by membrane permeabilization with 0.1% Triton X-100 (Sigma, Burlington, MA, USA) in PBS for 10 min at room temperature. Fixed cells were blocked using 5% (*w/v*) bovine serum albumin (BSA, Sigma, USA) in PBS for 1 h followed by incubation with PDCoV polyclonal antibody (prepared in our lab, 1:100 dilution) for 1 h. After washing three times with PBS, cells were incubated with fluorescent-labeled polyclonal goat anti-pig IgG antibody (Sigma, 1:100 dilution) for 1 h. Cells were washed 5 times with PBS, and further counterstained with DAPI (Sigma, Burlington, MA, USA) for another 5 min at room temperature. After being washed 5 times with cold PBS, the culture plate was observed under a fluorescence microscope (Zeiss, Oberkochen, Germany).

### 2.7. Immunoblotting Analysis

Protein lysates were obtained from ST cells using ice-cold lysis RIPA buffer containing 10 mM phenylmethylsulfonyl fluoride (PMSF, Solarbio, Beijing, China). Total protein concentration was determined by BCA protein assay kit (Beyotime, Shanghai, China). Equal amounts of protein were subjected to SDS-PAGE analysis and transferred onto polyvinylidene fluoride (PVDF) membrane (Millipore, Burlington, MA, USA). Membranes were blocked with 5% (*w/v*) nonfat milk for 3 h at room temperature and then incubated with primary antibody at 4 °C overnight. The following primary antibodies were used in the current study: β-actin (Abcam, Waltham, MA, USA) and PDCoV N monoclonal antibody (prepared in our laboratory using standard methods). After incubation with the primary antibody, the HRP-conjugated secondary antibody was added, and the solution was incubated for 2 h. Immunoblotting results were visualized with Luminata Crescendo Western HRP Substrate (Millipore, Burlington, MA, USA) on a GE AI600 imaging system and analyzed by ImageJ software.

### 2.8. Virus Titration by a Plaque Assay

ST cells in 6-well plates were used for all plaque assays for PDCoV, TGEV, and VSV-GFP propagated in both ST cells and LLC-PK cells. Virus titer was determined using the plaque assay as described previously [4]. It is worth noting that TGEV and VSV-GFP are measured without the addition of trypsin to the formulation.

### 2.9. Atomic Force Microscopy (AFM)

AFM analysis was conducted with an MFP3D Infinity-Asylum Research AFM in tapping mode (Oxford Instruments PLC, Oxfordshire, UK). Briefly, cells were seeded onto the 10 mm glass coverslip and cultured in a 6-well culture plate for 24 h. Subsequently, cells were inoculated with PDCoV (MOI = 10) and incubated at 4 °C for 1 h. After washing, cells were fixed with 4% formaldehyde in PBS. Imaging was performed with uncoated silicon cantilevers AC160TS-R3 from Oxford Instruments PLC, with a tip radius of 7 nm, resonance frequency of approximately 200–300 kHz, and a spring constant of 8.4–57 k (N/m). Images with a scan size of 1 × 1 μm2/4.5 × 4.5 μm^2^/20 × 20 μm^2^ and resolution 512 × 512 pixels^2^ were obtained with scan rates between 0.6 and 1.0 Hz and set points close to 0.2 V. AFM images were analyzed offline in AFM software (Ergo; Oxford Instruments PLC, Oxfordshire, UK).

### 2.10. Immunohistochemistry and Immunohistofluorescence

The jejunum, ileum, cecum, and colon of piglets at 3 dpi used in the experiment were taken from our previous artificial PDCoV-infection experiment. The distribution of PDCoV in tissues was determined using immunohistochemistry as described previously [19].

For SA staining, FITC-conjugated wheat germ agglutinin lectin (WGA, Sigma, Burlington, MA, USA) was used to stain total SA. A final concentration of 20 µg/mL of the WGA was added into the fixed tissues and incubated for 1 h at room temperature. The nuclei were stained with DAPI (Sigma, Burlington, MA, USA) and the fluorescence images were captured using the DS-U3 imaging system (Nikon, Tokyo, Japan).

### 2.11. Inhibition of Trypsin Activity by Aprotinin Assay

Aprotinin (Sigma, Burlington, MA, USA) was dissolved in DPBS to a final concentration of 2 mg/mL, filtered using a 0.22 μm membrane filters, and stored at 4 °C. The aprotinin was added to the trypsin solution to a final concentration of 2 μg/mL of aprotinin and then shaken at 37 °C for 30 min to inactivate the trypsin.

### 2.12. Statistical Analysis

Results were expressed as the means ± standard deviation (SD) from three independent experiments. Statistical analyses were performed using Student’s *t*-test. Differences were considered significant at *p* < 0.05. Statistical significance is indicated in figures as follows: * 0.01 < *p* < 0.05, ** *p* < 0.01.

## 3. Results

### 3.1. Removal of Surface Carbohydrate from ST/LLC-PK1 Cells by NaIO_4_ Significantly Reduced the PDCoV Infectivity to the Cells

To determine whether carbohydrate on the cell surface was essential for PDCoV infection, carbohydrate moieties were removed from ST and LLC-PK1 cells by treatment with NaIO_4_, which destroys carbohydrate groups without altering the cellular proteins or membranes [20]. Firstly, the effect of NaIO_4_ on cell proliferation was determined. The results showed that NaIO_4_ exhibited no significant cytotoxicity at concentrations from 0.2 to 5 mM on ST and LLC-PK1 cells (Appendix A). Then, we tested whether NaIO_4_ can alleviate the susceptibility of ST and LLC-PK1 cells to PDCoV. Here, TGEV, which could utilize carbohydrate moiety-SA as a receptor, was used as a positive control, and VSV-GFP was used as a negative control. ST/LLC-PK1 cells were pretreated with NaIO_4_ (0, 0.2, 1, and 5 mM) for 30 min at 4 °C and then infected with PDCoV (MOI = 0.1), TGEV (MOI = 0.1), and VSV-GFP (MOI = 0.1). Cells were fixed and tested by immunofluorescence assay at 8 hpi, or samples were collected at 24 hpi and assayed by RT-qPCR or TCID_50_. In the VSV group, no obvious effects were observed on the VSV-GFP replication when different concentrations of NaIO_4_ were added (Figure 1). These data demonstrate that the removal of cell surface carbohydrates does not affect VSV-GFP infection. But in the PDCoV and TGEV groups, the RT-qPCR results demonstrated that virus replication was significantly inhibited by adding 1 and 5 mM of NaIO_4_ (Figure 1A,C). The viral infectious titers of PDCoV and TGEV in the NaIO_4_ treated groups were markedly reduced when compared with those of the virus only groups (Figure 1B,D). Moreover, the NaIO_4_-treated PDCoV groups showed a more obvious inhibition effect on virus replication when compared with the NaIO_4_-treated TGEV groups. We observed the same phenomenon by IFA in both ST and LLC-PK1 cell lines (Figure 1E). These data indicated that carbohydrate moieties were required for the infection of PDCoV.

### 3.2. Sialic Acids Act as Receptors for PDCoV

SA is an abundant carbohydrate moiety on the cell surface and acts as a receptor for many viruses, especially for some coronaviruses. So the effect of SA on PDCoV infection was tested in the next experiment. To evaluate the role of SA during PDCoV infection, we used NA to remove cell surface SA prior to virus infection. Firstly, the cytotoxicity of NA was evaluated on ST and LLC-PK1 cells by CCK-8 assay. The results showed that NA exhibited no significant cytotoxicity at the concentrations from 0.0625 to 16 U (Appendix A). ST/LLC-PK1 cells were pretreated with NA (0, 0.0625, 0.25, 1, and 4 U) for 2 h at 37 °C and then infected with PDCoV (MOI = 0.1), TGEV (MOI = 0.1), and VSV-GFP (MOI = 0.1). After 24 h, the test was performed with qRT-PCR and TCID_50_. By RT-qPCR, it was found that pretreatment of cells with NA significantly reduced the infection of PDCoV in a dose-dependent manner, which is similar to the result of positive control of TGEV (Figure 2A,C). The TCID_50_ titers of PDCoV and TGEV in the cells of the NA-treated groups were obviously reduced when compared with the virus only groups, but there was no significant difference in the VSV-GFP group (Figure 2B,D). Similar results were obtained by IFA in both ST and LLC-PK1 cells (Figure 2E). Collectively, these results demonstrated that SA may play an important role during PDCoV infection.

### 3.3. Cell Surface SA Facilitates PDCoV Attachment

To characterize the mechanistic role of SA during PDCoV infection, NA-treated ST or LLC-PK1 cells were incubated with PDCoV (MOI = 10) at 4 °C for 2 h to promote virus attachment, and the attached virus was counted by plague assay, Western Blot (WB), and qRT-PCR. Our data indicated that the NA-treated cells significantly reduced PDCoV (Figure 3A,D) and TGEV (Figure 3B,E) attachment at 4 °C, while there was no difference in VSV-GFP particles attached to the NA-treated and untreated cells (Figure 3C,F). WB analysis showed that the PDCoV N protein expression was reduced approximately 20% in ST and 70% in LLC-PK1 cells with 4 U of NA treatment, respectively (Figure 3G). By RT-qPCR, it was found that pretreatment of cells with NA significantly reduced PDCoV attachment at 4 °C, which is similar to the result of WB (Figure 3G,H). It is noteworthy that the removal of SA seems to affect the attachment of PDCoV to LLC-PK1 cells more than that of ST cells, but the exact reason is unclear. Taken together, our findings provide direct evidence that SA on the surface of ST and LLC-PK1 cells can be used as an attachment receptor by PDCoV and thereby increase infection efficiency.

### 3.4. Detection of PDCoV Attachment to Cells by Atomic Force Microscopy

We further explored the effect of cell surface SA on PDCoV attachment by AFM. ST and LLC-PK1 cells were incubated with PDCoV (MOI = 10) at 4 °C to allow PDCoV to attach on the cell surface but did not enter the cells. AFM deflection images (Figure 4B,D) and 3D images (Figure 4A,C) of PDCoV were used to assess the number of PDCoV attached on the cell surface. The surface of the control cells was smooth, while the surface of cells incubated with PDCoV had many 130–180 nm particles, which was exactly the size of PDCoV particle. In contrast, viral particles on the NA-pretreated cell surface were significantly reduced compared to those NA-untreated cells, indicating that the attachment ability of PDCoV to the cell surface was decreased after the inhibition of SA.

### 3.5. Co-localization of PDCoV and SA in the Intestine of Piglets

To clarify whether PDCoV could recognize SA in vivo, the intestinal tissues from different parts (jejunum, ileum, cecum, and colon) of PDCoV-infected piglets were labeled using PDCoV N monoclonal antibody and SA using FITC-WGA, respectively. Clinically, PDCoV also mainly infects the small intestine of piglets, which is consistent with our previous results [19,21]. The distribution of PDCoV and SA were closely correlated, and SA was abundant on the surface of the small intestinal tissues (jejunum and ileum) (Figure 5A,B), where PDCoV was also present in large amounts. On the surface of the large intestinal tissues (cecum and colon) (Figure 5C,D), SA was less abundant and PDCoV was less capable of infection.

### 3.6. Trypsin Promotes PDCoV to Acquire Binding SA Activity

The swine small intestine contains a large amount of trypsin, and the addition of exogenous trypsin can also promote PDCoV infection in cell culture. Our previous report showed that PDCoV requires trypsin treatment to acquire the ability to agglutinate rabbit erythrocytes and that the ability of PDCoV to agglutinate rabbit erythrocytes can be inhibited by the NA treatment of erythrocytes [18]. So in the current experiment, we further investigated the interaction between exogenous trypsin and the cellular SA during PDCoV infection. We first prepared PDCoV without exogenous trypsin during propagation (namely PDCoV^T-^). Then the level of PDCoV^T-^ attached on ST and LLC-PK1 cells with or without NA treatment was detected by plaque assay. These results showed that treatment with NA could not reduce the attachment of PDCoV^T-^, and by comparison we found that PDCoV^T-^ seemed to lose the SA-binding ability (Figure 6A,D). In contrast, the viral titer of PDCoV^T-^ with trypsin treatment decreased 0.9 × 10^4^ PFU/mL and 3.76 × 10^4^ PFU/mL after ST and LLC-PK1 cells were treated with NA, respectively (Figure 6B,E). To determine whether the enzymatic activity of trypsin could affect PDCoV infection, we pre-incubated trypsin with its inhibitor, aprotinin (Sigma, 2 μg/mL), at 37 °C for 30 min to completely disrupt its enzymatic activity. After treatment of PDCoV^T-^ with inactivated trypsin, the results showed no significant difference in virus titers (Figure 6C,F). The results of WB also showed that NA treatment reduced PDCoV infection both in ST and LLC-PK1 cells (Figure 6H). But there were no significant changes in PDCoV^T-^ and PDCoV^T-^+trypsin- groups (Figure 6G,I). This result also indicated that the SA-binding activity of PDCoV was caused by the enzymatic activity of trypsin.

## 4. Discussion

It is reported that viruses bind with the glycocalyx first during viral infection, so several viruses employ the glycocalyx as their attachment receptor to facilitate infection [10]. The interaction between glycocalyx and the CoV spike(S) protein, which is responsible for receptor recognition, has also been demonstrated in coronaviruses. For example, SARS-CoV, SARS-CoV-2, and HCoV-NL63 are able to bind to HS [11,22,23]; HCoV-OC43 and BCoV bind to 5-N-acetyl-9-O-acetyl- neuraminic acid [24,25]; MERS-CoV binds to 5-N-acetyl-neuraminic acid, and guinea fowl coronavirus binds to biantennary di-N-acetyllactosamine or SA-capped glycans [26,27]. It is proved that TGEV, PEDV, FeCV, and IBV can also bind to the SA to promote their infection efficiency, although the type of SA they recognize is still unknown.

In this study, we demonstrated that PDCoV also has the ability to recognize cellular glycan receptors, and it uses SA as an attachment receptor. However, SA is not essential during PDCoV infection, which might be due to the dependence of PDCoV on other protein receptors. This phenomenon has also been found in other coronaviruses that can bind to both SA and protein receptors, such as MERS-CoV, TGEV, and mouse hepatitis virus (MHV), which use dipeptidyl peptidase 4 (DPP4), pAPN, and carcinoembryonic antigen cell adhesion molecule 1 (CEACAM1a) as receptors, respectively [14,16,28,29,30]. The removal of SA could not block the infection of these viruses completely. Thus, we speculated that the binding of PDCoV and SA requires other protein receptors during the infection. Studies on HS, another common glycoconjugate receptor, have found that in addition to promoting viral adhesion, HS can promote the binding of the S protein of SARS-CoV-2 to ACE2 significantly by enhancing the open conformation of the receptor binding domain of SARS-CoV-2 [11]. However, it is still unconfirmed whether SA would play a similar role as HS, which deserves further investigation.

The receptor-binding capability of the virus and the distribution of receptors are critical for the host range, tissue tropism, and pathogenesis of the virus. Both PDCoV and TGEV mainly infect the intestine and use pAPN and SA as their receptor or attachment receptor [5,28], while the TGEV S protein N-terminal deletion strain porcine respiratory coronavirus (PRCoV) mainly infects the respiratory tract because the lack of binding activity to SA of its S protein might be involved in its tissue tropism [31]. Based on this, we studied the distribution of SA and PDCoV in different tissues of PDCoV-infected piglets, and we found that the distribution of PDCoV and SA showed a positive correlation only in intestinal tissues, but there was no significant correlation in other tissues. pAPN are widely expressed in various tissues, so it is also inappropriate to explain the tissue tropism of PDCoV from the perspective of pAPN.

Trypsin is rich in the intestine compared to other tissues, and PDCoV requires the addition of exogenous trypsin to mimic the intestinal environment during isolation and passage [4]. The previous study in our laboratory found that PDCoV could agglutinate rabbit erythrocytes only after treatment with trypsin, demonstrating that trypsin-treated PDCoV acquired SA-binding activity [18]. In this study, we further demonstrated that trypsin treatment induced PDCoV to acquire SA-binding activity. We suggest that this phenomenon may avoide the binding of the progeny virions to the surface SA of infected cells during the release stage. For example, influenza viruses promote virion release from the cell surface by sialidase NA (IAV, IBV) or sialate-O-acetyl-esterases (ICV) [32,33,34]. But coronaviruses lack sia-destroying enzymes, and it is widely believed that the reversible interaction of the virus with SA is controlled only by the binding equilibrium [35]. Based on our data, we hypothesize that, when the PDCoV replication in cells finishes, it could not bind to SA efficiently because of the absence of intracellular trypsin, thus making it easier to release rather than to bind it on the infected cell surface. When PDCoV is released into the intestine, it acquires SA-binding activity due to the digestion by trypsin, which is widely distributed in the intestine, making it easier for PDCoV to attach to un-infected cells in the complex conditions of the intestine and thus, to promote infection.

Cross-species transmission is common among coronaviruses. For example, SARS-CoV-2 and SARS-CoV, which have caused great harm to humans, may originate from bats, and MERS-CoV may originate from camels [36,37,38]. Studies related to their cross-species transmission have mainly focused on their corresponding protein receptors, and similarly for PDCoV, studies on different species of APN seem to shed some light on the reasons for its cross-species transmission [5]. For influenza viruses, SA-binding preference is a key determinant of its host. Sialic acid α2–3 galactose (SAα2,3Gal) distributed predominately on poultry cell surface is the main recognition and binding site for avian influenza viruses, whereas human influenza viruses mainly recognize and bind to α2–6 galactose receptors (SAα2,6Gal) on the host cell surface [39]. The binding of influenza virus particles to receptors on the cell surface is an indispensable and critical step for virus infection, and therefore avian influenza virus is not easily transmitted from person to person [40]. In pigs, two types of receptors, SAα2,3Gal and SAα2,6Gal, are widely distributed, so pigs can facilitate the ability of the avian influenza virus to infect humans by act as a “mixer” [41,42,43]. PDCoV originated from birds and gradually adapted and spread widely in pigs, and there is clear evidence that PDCoV can infect humans [9,44]. Therefore, whether SA as its co-receptor plays a facilitating role in the cross-species transmission of PDCoV and whether pigs play a bridging role in the spread of PDCoV from birds to humans deserve further investigation.

## Figures and Tables

**Figure 1 viruses-13-02442-f001:**
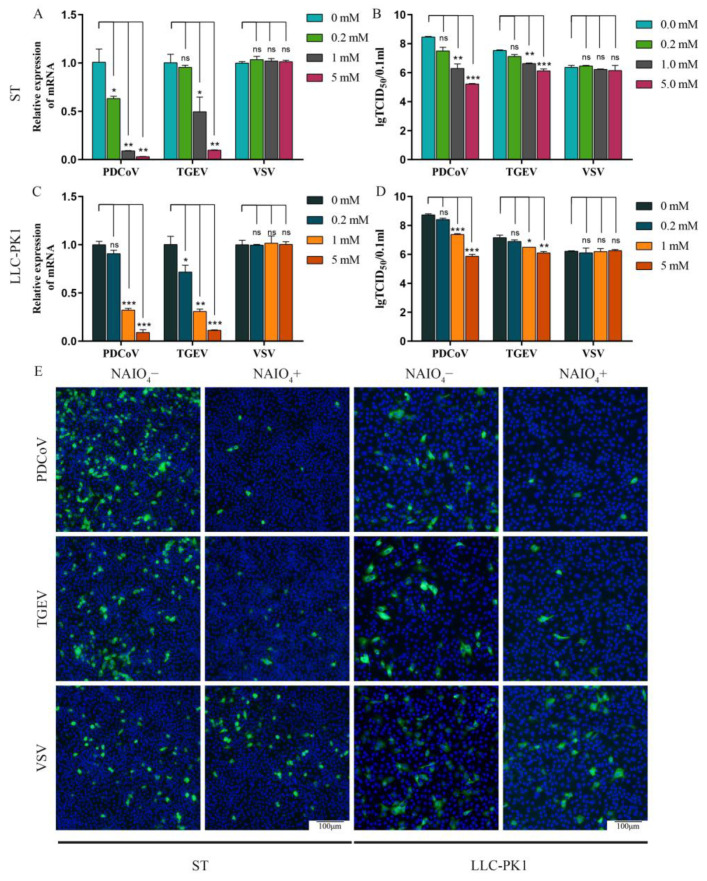
NaIO_4_ inhibits PDCoV infection at ST and LLC-PK1 cell. ST/LLC-PK1 cells were pretreated with NaIO_4_ for 30 min and infected with PDCoV, TGEV, and VSV. Samples were collected at 24 hpi and assayed by RT-qPCR (**A**,**C**), TCID_50_ (**B**,**D**), or cells were fixed and tested by immunofluorescence assay (**E**) at 8 hpi. Experiments were performed at least three times. Differences were considered significant at * *p* < 0.05, ** *p* < 0.01, *** *p* < 0.001; ns: differences were not significant; one-way ANOVA.

**Figure 2 viruses-13-02442-f002:**
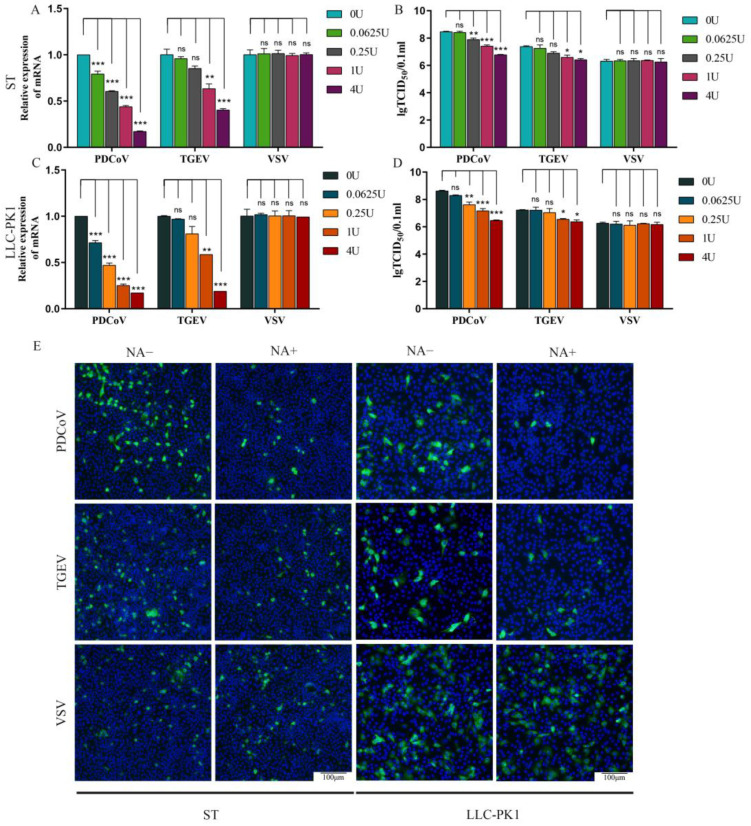
NA inhibits PDCoV infection at ST and LLC-PK1 cell. ST/LLC-PK1 cells were pretreated with NA for 2 h and infected with PDCoV, TGEV, and VSV. Samples were collected at 24 hpi and assayed by RT-qPCR (**A**,**C**), TCID_50_ (**B**,**D**), or cells were fixed and tested by immunofluorescence assay (**E**) at 8 hpi. Experiments were performed at least three times. Differences were considered significant at * *p* < 0.05, ** *p* < 0.01, *** *p* < 0.001; ns: differences were not significant; one-way ANOVA.

**Figure 3 viruses-13-02442-f003:**
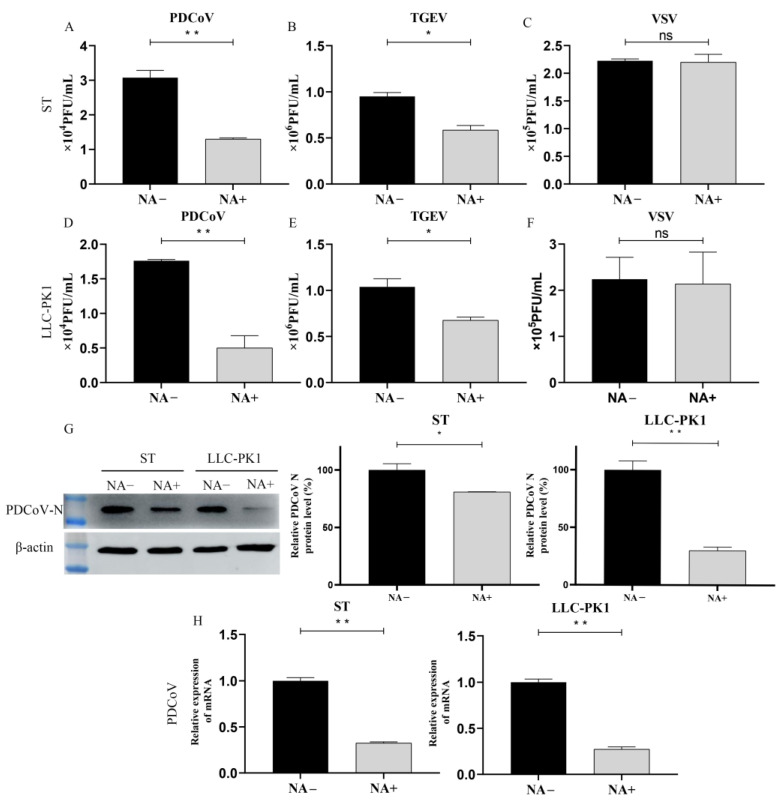
PDCoV binding requires sialic acid. ST/LLC-PK1 cells were pretreated with NA for 2 h and incubated with PDCoV (**A**,**D**), TGEV (**B**,**E**), and VSV (**C**,**F**) at 4 °C to promote virus attachment. After attachment, samples were collected immediately after 3 washes with DPBS. Viral replication was estimated by plague assay, WB (**G**), and qRT-PCR (**H**) at ST and LLC-PK1 cell. Experiments were performed at least three times. Differences were considered significant at * *p* < 0.05, ** *p* < 0.01; ns: differences were not significant; *t*-test.

**Figure 4 viruses-13-02442-f004:**
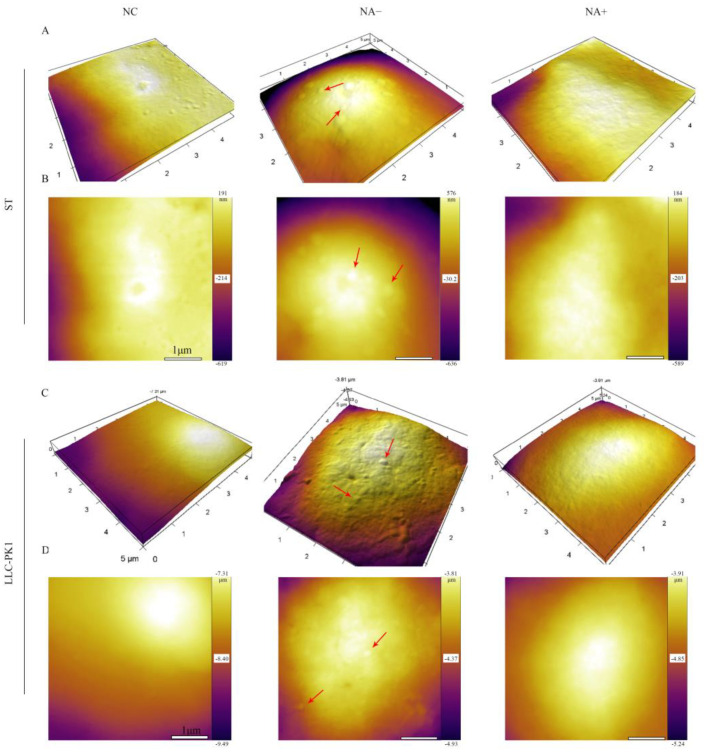
Detection of PDCoV adhesion to ST and LLC-PK1 cells by AFM.ST/LLC-PK1 cells were pretreated with NA for 2 h and incubated with PDCoV at 4 °C to promote virus attachment. AFM deflection images (**B**,**D**) and 3D AFM images (**A**,**C**) of PDCoV adsorbed on the cell surface. Those figures show high-resolution images recorded in the square regions. PDCoV adsorbed on the cell surface is indicated by a red arrow.

**Figure 5 viruses-13-02442-f005:**
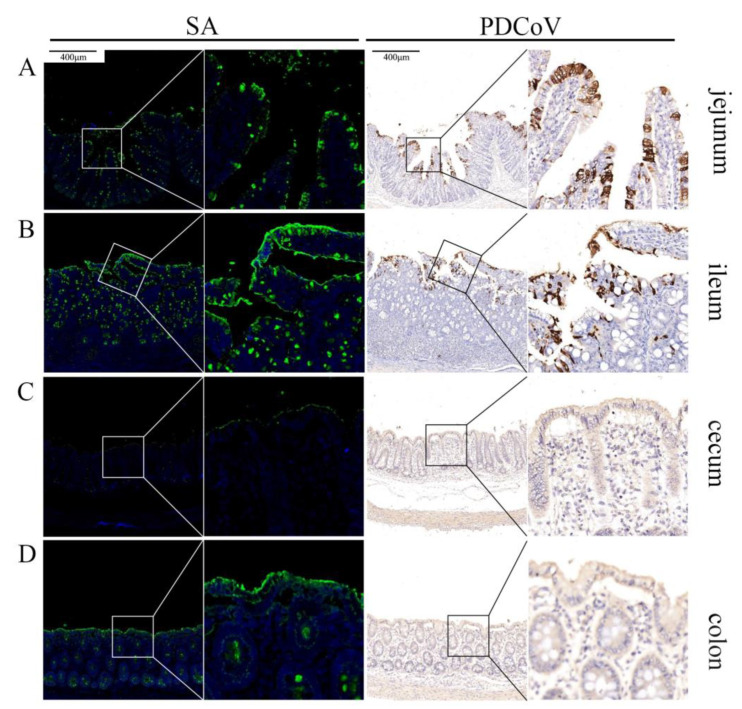
Co-localization of PDCoV and sialic acid in the intestine of piglets. PDCoV-infected piglets show strongly PDCoV positive in the jejunum (**A**) and the ileum (**B**) and negative in the cecum (**C**) and the colon (**D**). SA is widely distributed in the jejunum (**A**) and the ileum (**B**), but less distributed in the cecum (**C**) and the colon (**D**).

**Figure 6 viruses-13-02442-f006:**
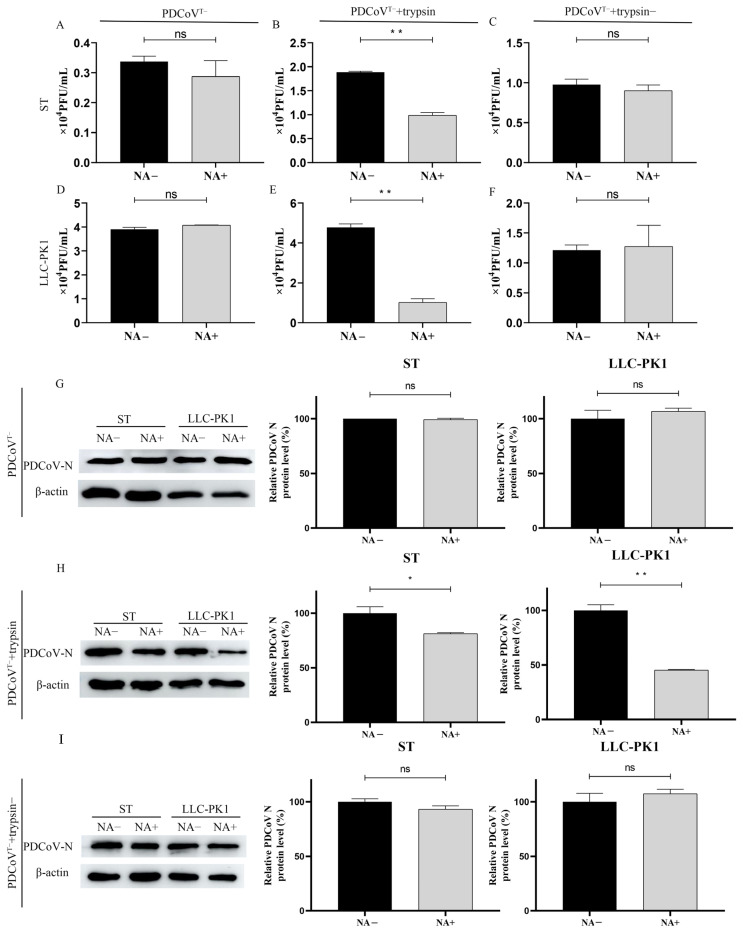
Trypsin promotes PDCoV to acquire agglutinating sialic acid activity. We detected the level of PDCoV^T-^ (**A**,**D**,**G**), PDCoV^T-^+trypsin (**B**,**E**,**H**), and PDCoV^T-^+trypsin- (**C**,**F**,**I**) attached on ST cells and LLC-PK1 cells before and after NA treatment by plaque assay and WB. Differences were considered significant at * *p* < 0.05, ** *p* < 0.01; ns: differences were not significant; *t*-test.

## Data Availability

The data used to support the findings of this study are available from the corresponding author upon reasonable request.

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
