# Peer review of "Porcine Deltacoronavirus Utilizes Sialic Acid as an Attachment Receptor and Trypsin Can Influence the Binding Activity"

_viruses, 2021, doi:10.3390/v13122442_

Round 1

Reviewer 1 Report

In this manuscript, Yuan et al., conducted a study aimed to confirm sialic acid (SA) as an attachment receptor for PDCoV invasion and further found an essential role of trypsin on the PDCoV-SA binding. This is an important topic as key functional receptor for PDCoV has not been determined until now, except for a known entry receptor APN for PDCoV.

The authors used several approaches to investigate the virus binding and infection by measuring viral RNA (by qPCR), N protein expression (by Western blot) and virus titers (by TCID50 or plaque assay). The authors concluded that sialic acid (SA) is a novel attachment receptor for PDCoV infection and their binding is dependent on the pre-treatment of trypsin. Overall, even if the subject is interesting, some weaknesses need to be improved.

(1) The expression levels of sialic acid (SA) on ST and LLC-PK1 cells before and after drugs treatment should be determined by experiments. It may explain the author's question that the removal of SA seems to affect the attachment of PDCoV to LLC-PK1 cells stronger than that of ST cells.

(2) PDCoV has higher virus titers than that of TGEV and VSV in Figure 1 and 2, but has the lowest virus titer in Figure 3. Please explain.

(3) Line 304 Page 10: The author concluded that PDCoVT- seemed to lose the SA binding ability based on the results of Figure 6A and D. However, no direct experimental results supported this conclusion.

(4) The experimental detail that antitrypsin can completely disrupt enzymatic activity of trypsin at 37°C for 30 min should be described and the data should be provided.

(5) Line 313 Page 11: replace "Fig.6 G and E" by " Fig.6 G and I "

(6) PDCoV seems to have a higher infection rate in ST cells than that in LLC-PK1 cells based on the results of Figure 1E and 2E; however, obvious reverse results were observed in Figure 6A, B, D and E. Please explain.

(7) The results of CCK8 assay in Line 195 Page 5 and Line 225 Page 7 should be provided as supplementary materials.

(8) NaIO4 in Line 65, 92, 97,106 should be NaIO4 and CO2 in Line 78 should be CO2. Line 176, ‘[19]’ should be placed before the point.

(9) Reference should include author’s name, title of the article, abbreviated Journal name, year, volume, and page range. Please correct them and carefully check the accuracy of the names.

(10) The authors found that the distribution of PDCoV and SA showed a positive correlation only in intestinal tissues, but there was no significant correlation in other tissues. However, the author selected two non-intestinal epithelial cells, ST and LLC-PK1 cells, to perform in vitro studies. Some key experiments in Figure 1, 2, 3, and 6 should be performed in a cell line of more physiologically relevant intestinal epithelial cells, such as IPI-2I or IPEC-J2 cells. 

Reviewer 2 Report

Yuan et. al reported PDCoV utilized sialic acid as an attachment receptor through treatment with NaIO4 or neuraminidase. And sialic acid and PDCoV have similar distribution in swine intestine. Furthermore, the trypsin activation is also essential for the utilization of sialic acid by PDCoV. The overall study is interesting. And the experiments were well-designed and performed. The finding will add to the field of PDCoV entry as well as pathology in swine. I have only several minor comments.

  1. For the binding assay, besides the plaque assay, measuring the binding viral genome RNA by qPCR is widely used. This quantification method will further confirm salic acid participate the attachment process. AFM assay looks good. But it is not a quantification method. Since this is the key experiment to support the main conclusion. I suggest adding this line of evidence.

  1. The distribution of PDCoV and sialic acid is similar in swine intestine. This experiment is very important in this study. It indicates the significance of sialic acid in physiological conditions. My suggestion to provide more details about this part. For example, is it possible to provide a merged picture for SA and PDCoV N? It will be perfect if there is a colocalization analysis using multiple pictures.

  1. The relationship between pAPN and SA. As the authors mentioned, pAPN might not the sole receptor utilized by PDCoV. Will the virus utilize pAPN when SA is not available, or SA is indispensable on pAPN?

  1. The scale bar is missing in all of the pictures including figure 1E, 5A-D.

Author Response

Reviewer

Question 1:For the binding assay, besides the plaque assay, measuring the binding viral genome RNA by qPCR is widely used. This quantification method will further confirm salic acid participate the attachment process. AFM assay looks good. But it is not a quantification method. Since this is the key experiment to support the main conclusion. I suggest adding this line of evidence.

 Response:Thank you for your comments, we have added this part of the qPCR experiment results and description to new manuscript according to your suggestion.

Question 2:The distribution of PDCoV and sialic acid is similar in swine intestine. This experiment is very important in this study. It indicates the significance of sialic acid in physiological conditions. My suggestion to provide more details about this part. For example, is it possible to provide a merged picture for SA and PDCoV N? It will be perfect if there is a colocalization analysis using multiple pictures. 

Response:Thank you for your suggestion. We have tried to carry out the immunohistofluorescence assay using PDCoV N antibody for several times. However, the images did not clear because of the N antibody applicability. Our previous data showed that the monoclonal antibody for PDCoV N protein cannot be used for immunohistofluorescence detection but suitable for immunohistochemistry assay. So we used two seperatWe next to prepare more monoclonal antibodies of PDCoV that can be used for immunohistofluorescence assay. We will then perform these experiments to provide more details of PDCoV and sialic acid distribution.

Question 3:The relationship between pAPN and SA. As the authors mentioned, pAPN might not the sole receptor utilized by PDCoV. Will the virus utilize pAPN when SA is not available, or SA is indispensable on pAPN?

 Response:Thanks for your helpful comments. We have done some relevant studies on pAPN and we next plan to study the relationship between SA and pAPN. We plan to establish knockout cell lines for pAPN and SA and co-knockout cell lines respectively for the next verified. To explore the respective roles of pAPN and SA during PDCoV infection and whether these two potential receptors would jointly involved in PDCoV infection.

Question 4:The scale bar is missing in all of the pictures including figure 1E, 5A-D.

Response:Thanks for your helpful comments. We have marked the scale bar of Figure 1E in the lower right corner of the second and fourth columns, and the scale bar of Figure 5 is marked in the upper left corner of the first and third columns.

Reviewer 3 Report

Yuan et al. confirmed that porcine deltacoronavirus (PDCoV) could utilize sialic acid (SA) on the surface of swine testicular (ST) and LLC porcine kidney (LLC-PK1) cells as an attachment receptor and trypsin treatment could increase the SA-binding activity of PDCoV. The study designs were scientific and reasonable. The data were convincing, providing novel evidence for understanding the mechanism of PDCoV infection. 

Minor comments:

1. In the study, NaIO4 and NA treatment were for cells (ST and LLC-PK1), not for PDCoV. Thus, the several statements in the text should be modified, such as “alleviate PDCoV infection”, “inhibit PDCoV infection” in the abstract, and “PDCoV infection” in line 189, “NaIO4 can inhibit PDCoV infection on ST and LLC-PK1 cells” in line 196, “a more obvious inhibition effect on virus replication” in line 209 in the results section. Theses can be replaced by alternative descriptions, e.g. “alleviate the susceptibility of cells to PDCoV”, or “influence the PDCoV infectivity to the cells”.

2. In line 202, what does the “antiviral effects” mean here? This needs to be revised.

3. The GenBank no or respective reference regarding “The TGEV HN-2012 strain used in the study should be provided.

4. The English writing of the manuscript needs to be further improved.

Author Response

Reviewer 

Question 1:In the study, NaIO4 and NA treatment were for cells (ST and LLC-PK1), not for PDCoV. Thus, the several statements in the text should be modified, such as “alleviate PDCoV infection”, “inhibit PDCoV infection” in the abstract, and “PDCoV infection” in line 189, “NaIO4 can inhibit PDCoV infection on ST and LLC-PK1 cells” in line 196, “a more obvious inhibition effect on virus replication” in line 209 in the results section. Theses can be replaced by alternative descriptions, e.g. “alleviate the susceptibility of cells to PDCoV”, or “influence the PDCoV infectivity to the cells”.

Response:Thanks for your helpful comments. We have revised the sentence according to your suggestion.

Question 2:In line 202, what does the “antiviral effects” mean here? This needs to be revised.

Response:Thanks for your helpful suggestions. The sentence has been modified to“In the VSV group, no obvious effects were observed on the VSV-GFP replication when different concentrations of NaIO4 were added (Fig.1).”

Question 3:The GenBank no or respective reference regarding “The TGEV HN-2012 strain used in the study should be provided.

Response:Thanks for your helpful comments. TGEV HN-2012 strain was isolated and identified in our laboratory, and serial propagated in PK-15 cells. The biological characteristics of this strain were investigated , and we sequenced the complete genomes of TGEV HN-2012 strain. Our sequence of TGEV-HN2012 strain will be uploaded to the GenBank later and we can submit the GenBank ID to the editorial team after we have completed our submission if required.

Question 4:The English writing of the manuscript needs to be further improved.

Response:Thanks for your helpful comments. We have carefully reviewed and revised the manuscript. Moreover, our manuscript also has been reviewed and edited by a professional English writer and the corresponding descriptions have revised in the new version of manuscript.